# Nuclear Magnetic Resonance Dynamics of LiTFSI–Pyrazole Eutectic Solvents

**DOI:** 10.3390/ma18225184

**Published:** 2025-11-14

**Authors:** Emilia Pelegano-Titmuss, Muhammad Zulqarnain Arif, Giselle de Araujo Lima e Souza, Phillip Stallworth, Yong Zhang, Adam Imel, Thomas Zawodzinski, Steven Greenbaum

**Affiliations:** 1Department of Physics and Astronomy, CUNY Hunter College, New York, NY 10065, USA; emilia.peleganotitmuss45@myhunter.cuny.edu (E.P.-T.); pstallwo@hunter.cuny.edu (P.S.); sgreenba@hunter.cuny.edu (S.G.); 2Department of Chemical and Biomolecular Engineering, University of Tennessee, Knoxville, TN 37996, USA; marif1@vols.utk.edu (M.Z.A.); aimel@utk.edu (A.I.); tzawodzi@utk.edu (T.Z.); 3Department of Chemical and Biomolecular Engineering, University of Notre Dame, Notre Dame, IN 46556, USA; yzhang19@nd.edu; 4University of Tennessee-Oak Ridge Innovation Institute, Oak Ridge, TN 37830, USA

**Keywords:** FFC-NMR, diffusion NMR, deep eutectic solvents, lithium-ion batteries, electrolytes

## Abstract

Deep Eutectic Solvents (DESs) have emerged as promising candidates to replace conventional organic solvents in various technological applications due to their low vapor pressure, non-flammability, and ease of preparation at low costs. In particular, Type IV DESs, which are composed of metal salts and hydrogen bond donors, are possible replacements for lithium-ion battery electrolytes. In this study, we investigate the molecular dynamics of solvents of lithium bis(trifluoromethanesulfonyl)imide (LiTFSI) and pyrazole (PYR) at varying LiTFSI:PYR molar ratios (1:2, 1:3, 1:4, 1:5) using Nuclear Magnetic Resonance Dispersion (NMRD) and Pulsed Field Gradient (PFG) Nuclear Magnetic Resonance (NMR). PFG NMR reveals composition-dependent diffusion trends, while NMRD provides molecular-level insights into the longitudinal relaxation rate (R_1_ = 1/T_1_). Notably, the LiTFSI:PYR (1:2) sample shows distinct behavior across both techniques, exhibiting enhanced relaxation rates and lower self-diffusion for ^1^H compared to the other nuclei (^19^F and ^7^Li), suggestive of stronger and more efficient Li^+^–pyrazole interactions, as confirmed by the modeling of the relaxation profiles. Our study advances understanding of ion dynamics in azole-based eutectic solvents, supporting their potential use in safer battery electrolytes.

## 1. Introduction

Lithium-ion batteries (LIBs) are the most common battery type used in energy storage systems, due to their high energy and power densities, long life cycle, and fast recharge [1]. They have revolutionized our society by enabling the development of wireless technology, such as mobile phones and laptops, as well as electric transportation. The increasingly common use of electric vehicles, such as electric cars, shows that LIBs have the potential to be used as a clean technology and reduce CO_2_ emissions from the use of gas-powered vehicles [2]. As an energy storage system, LIBs can also contribute to increasing the reliability of intermittent renewable energy, such as wind and solar power, through storing the energy that these systems generate over time, as well as decreasing our reliance on fossil-fuel powered plants [2]. However, while LIBs have the potential to impact our environment positively, improvements are still needed, particularly in the battery electrolyte, in order to ensure the safety and eco-friendliness of this technology.

A typical LIB electrolyte consists of a solution of organic carbonates, usually ethylene carbonate (EC) or propylene carbonate (PC) and dimethyl carbonate (DMC) or diethyl carbonate (DEC), and an added lithium salt, which is most commonly LiPF_6,_ and Solid Electrolyte Interphase (SEI) film-forming additives [3]. However, these organic carbonates are flammable and derived from fossil fuels [4], which makes the production of LIBs detrimental to the environment, and a typical LIB electrolyte poses a threat of battery fires and explosions. LiPF_6_ was the chosen lithium salt for LIBs because it offers an efficient ionic transport mechanism, necessary for the battery to charge and discharge [3]. However, LiPF_6_ is thermally unstable and begins to decompose at temperatures around 60 °C to 80 °C [3]. This, combined with the flammability of the organic carbonates in typical Li-ion electrolytes, means that when the battery electrolyte is exposed to air, fire and explosions may occur [5]. The flammability and thermal instability of the organic carbonates in current electrolytes can also trigger thermal runaway [5], and thermal or mechanical abuse of the batteries can lead to the generation of heat, the starting of fires, the decomposition of LiPF_6_, and the release of toxic hydrofluoric acid [5,6]. These are major threats to human health and safety, and it is therefore necessary to develop a safer and more environmentally friendly LIB electrolyte.

Recent developments with ionic liquids have garnered attention in the scientific community for their potential to improve the safety and reliability of LIB electrolytes. This is due to their thermal stability, electrochemical stability, and low vapor pressure [7,8]. However, many ionic liquids are also derived from nonrenewable feedstocks and are very costly due to their time-consuming and laborious synthesis [9]. They also have other cations present besides Li^+^ that hinder the ionic mobility of the system due to the increase in electrostatic interactions [8], and therefore decrease the lithium transference number. As a result, other potential electrolyte replacements are being explored.

As an alternative to ionic liquids, Deep Eutectic Solvents (DESs) have emerged as a sustainable, innovative material for LIB electrolytes [10]. A eutectic mixture consists of two or more solid components that, when combined at a certain molar ratio, result in the lowest possible melting point of the mixture, which is a lower melting point than either of the components [11]. A DES is a eutectic mixture that has a deeper melting point than the ideal eutectic [11]. DESs have many benefits, including low toxicity, high biodegradability, and easy preparation with 100% atom-efficiency [11]. DESs are organized into five types based on these precursor materials, and of particular interest is the Type IV DES, which consists of a metal salt and a hydrogen bond donor [11].

The hypothesis that Type IV DESs containing azoles and LiTFSI could be a potential replacement for LIB electrolytes relies on the fact that they are non-flammable and highly conductive and can have favorable Li^+^ transference numbers [12]. In theory, the lithium-transference number of Type IV DESs is higher than that of ionic liquids because Type IV DESs only contain one type of cation (Li^+^), whereas ionic liquid electrolytes doped with lithium salt can have at least two cations (the ionic liquid cation and the Li^+^) [13]. A favorable lithium-transference number also can occur because of the expected coordination between the Li^+^ in the LiTFSI and the nitrogen in azoles, according to Lewis acid–base interactions, which favors a disassociation of the Li^+^ from the LiTFSI [12]. LiTFSI–Pyrazole (PYR) Type IV Eutectic Solvents (ESs) (see Figure 1) are of particular interest because they have the potential to achieve higher ionic conductivities, as it was found that LiTFSI:PYR (1:4) reached an ionic conductivity of 5.8 × 10^−3^ S cm^−1^ at 80 °C [12]. Additionally, LiTFSI:PYR (1:4) was found to have a stable electrochemical window and cycling performance [12].

While LiTFSI:PYR eutectic mixtures have shown promise as liquid electrolytes for lithium-ion batteries [12], their practical performance ultimately depends on how efficiently lithium ions are transported at the molecular scale. Therefore, it is essential to characterize the underlying molecular dynamics and, in particular, how they govern the interactions between Li^+^, pyrazole, and the TFSI^−^ anion. These microscopic processes directly influence macroscopic transport properties such as viscosity, ionic conductivity, and lithium transference number.

For this, Fast-Field Cycling (FFC) Nuclear Magnetic Resonance (NMR), unlike typical NMR instrumentation that measures relaxation at a fixed magnetic field, allows the measurement of longitudinal relaxation rate (R_1_ = 1/T_1_) at various magnetic field strength across a wide Larmor frequency range, going from a few kilohertz to several megahertz. FFC NMR produces a relaxation profile known as Nuclear Magnetic Resonance Dispersion (NMRD) [14,15].

Most importantly, FFC NMR is well-suited for evaluating the interplay between pyrazole and Li^+^ at a local level, as well as the potential for this Eutectic Solvent (ES) to transport lithium ions. This is because FFC NMR can distinguish between the behaviors of ^1^H, ^19^F, and ^7^Li individually and portray element-specific insight into how cations, anions, and solvent molecules move and interact. This level of detail cannot be captured by conventional spectroscopic techniques, which typically probe electronic or vibrational transitions [16] but do not resolve nuclear-scale translational and rotational dynamics across multiple frequency regimes.

By correlating these microscopic dynamics with macroscopic properties such as viscosity and ionic conductivity, this work delivers a comprehensive understanding of ion motion in LiTFSI–Pyrazole eutectic solvents. These fundamental insights support the rational design of greener, safer, and higher-performance liquid electrolytes with improved charge density and lithium transference number for next-generation lithium-ion batteries.

## 2. Materials and Methods

### 2.1. Sample Preparation

The eutectic solvents (ESs) were prepared by mixing pyrazole (PYR, CAS Number 288-13-1, Thermo Fisher Scientific (Walthman, MA, USA), purity 98%) and lithium bis(trifluoromethanesulfonyl)imide (LiTFSI, CAS Number 90076-65-6, Sigma-Aldrich (St. Louis, MA, USA), purity 99.99%) in specific molar ratios. Samples were prepared by fixing the molar ratio of LiTFSI at 1 while varying the molar ratio of pyrazole from 2 to 5. The four mixtures, LiTFSI:PYR (1:2–1:5), were placed on a hot plate and heated at 60 °C under stirring at 200 rpm until a homogeneous transparent liquid was obtained. After preparation, the samples were vacuum-dried for at least 24 h and transferred into an argon glovebox.

### 2.2. Diffusion NMR

Pulsed Field Gradient (PFG) NMR was employed to investigate the translational dynamics by measuring the self-diffusion coefficients of the four LiTFSI–Pyrazole samples. The samples were transferred to 5 mm glass NMR tubes inside an argon-filled glovebox and flame-sealed to prevent air and moisture exposure. Diffusion experiments were carried out at 10 °C using a model PXF4 temperature controller (Fuji Electric Co., Ltd., Tokyo, Japan), ensuring a temperature stability of ±1 °C. Nitrogen gas chilled in a dry-ice bath was used as the variable temperature (VT) medium.

Measurements were conducted on a 300 MHz NMR spectrometer (7T, Agilent/Varian DDR console, Santa Clara, CA, USA) equipped with a Doty Z-spec PFG probe for the nuclei ^1^H, ^19^F, and ^7^Li. The diffusion time (Δ) and gradient pulse duration (δ) were optimized between 20–100 ms and 2–5 ms, respectively. The maximum gradient strength applied was 850 G·cm^−1^. Raw spectra were manually phased and then baseline-corrected automatically. An exponential line broadening of 0.3 Hz was applied in the F2 dimension. Self-diffusion coefficients were extracted using the Stejskal–Tanner equation (Equation (1)) [17].(1)$$\frac{I}{I_{0}}=exp(-{D\gamma }^{2}g^{2}\delta^{2}\left(∆-\frac{\delta }{3}\right))\;$$

Here, *I* is the signal intensity with an applied gradient, *I*_0_ is the initial signal intensity, *γ* is the gyromagnetic ratio of the investigated nucleus, *g* is the gradient strength applied, Δ is the diffusion time, δ is the gradient pulse duration, and *D* is the self-diffusion coefficient.

### 2.3. Longitudinal Relaxation NMR

Fast Field Cycling NMR was utilized to investigate the ^1^H, ^19^F, and ^7^Li relaxation dynamics of the system as a function of magnetic field, ranging from 30 kHz to 15–35 MHz in ^1^H Larmor frequency. All measurements were performed at 10 °C to obtain a better dispersion profile in the NMRD analysis. The samples were transferred to 10 mm glass NMR tubes inside an argon-filled glovebox and flame-sealed, again to prevent air and moisture exposure. They were then measured using a Spinmaster FFC2000 CDC Relaxometer (Stelar, Italy) with either the standard pre-polarized (PP) or non-polarized (NP) pulse sequences, depending on the relaxation field [15]. For all measurements, the polarization field was set at 15 MHz for ^1^H and ^19^F in their respective probes, and 24 MHz for ^7^Li, with a field slew rate of 13 MHz·ms^−1^ and a switching time of 3 ms. Relaxation dispersion curves were obtained from the following relaxation field ranges for each nucleus: 30 kHz to 15 MHz for ^1^H, 28 kHz to 14 MHz for ^19^F, and 12 kHz to 12 MHz for ^7^Li. All the frequencies are expressed in terms of the nuclei Larmor frequency (*ν* = *γB*_0_/2*π*). The minimum and maximum relaxation fields for all measurements were determined based on the theoretical operating limits of the FFC NMR instrument SpinMaster FFC2000 (Stelar, Italy). Relaxation dispersion curves were based on 4–64 signal scans acquired for 8 delay (*τ*) values at each ionic species’ respective frequency. Finally, relaxation times (*T*_1_) were calculated from the magnetization recovery curves derived from the signal scans by fitting a mono-exponential equation to the data (Equation (2), Figure 2).*M_z_*(*τ*) = *M*_0_(*B_R_*) + [*M*_0_(*B_P_*) − *M*_0_(*B_R_*)] exp(*−τ*/*T*_1_(*B_R_*)) (2)

Here, *B_R_* is the relaxation field, *B_P_* is the polarization field, *M_z_* is the magnetization as a function of the delay time (*τ*), *M*_0_ is the initial magnetization, and *T*_1_ is the relaxation time. Equation (2) is derived from the Bloch equation and Curie’s law for magnetization [18].

T_1_ values were also measured using other spectrometers (90–500 MHz) to extend the NMRD profile over a broader range of relaxation fields. These included a fixed-field 300 MHz Varian spectrometer, 400 MHz and 500 MHz Bruker spectrometers, and a permanent magnet 90 MHz from Anasazi Instruments, with the latter applied only to the ^1^H and ^19^F nuclei of the LiTFSI:PYR (1:2) sample. The inversion-recovery pulse sequence was used for all fixed field experiments, and raw data were acquired with a list of 16 delay (*τ*) values. Relaxation times (T_1_) were then obtained by fitting the data using the *T*_1_*/T*_2_ module in TopSpin for the Bruker spectrometers (Bruker Corporation, Ettlingen, Germany) and OriginLab 2024 Version 10.1.0.178 for the Varian and Anasazi Instruments permanent magnets.

### 2.4. Relaxation Models

Quite a few studies have investigated the relaxation of eutectic solvents containing ions [19,20,21,22,23,24,25,26]. However, the use of quantitative models to study the relaxometry of eutectic solvents is in its early stages, with only a few studies performing these analyses on eutectic solvents [21,23]. These quantitative models are developed through the analysis of molecular-level interactions within the eutectic solvent. For dipolar nuclei with spins 1/2 (I = 1/2), such as ^1^H and ^19^F, the longitudinal relaxation is caused by fluctuations in the local magnetic field driven by intra- and intermolecular interactions [27]. More specifically, the relaxation at a nuclear site is governed by the local dynamics of fluctuating dipole–dipole interactions, whose strength, timescale, and correlation mechanisms determine the spectral density at the Larmor frequency. These fluctuations enable energy exchange between the spin system and the lattice, thereby defining the T_1_ relaxation pathway [27]. Note that homonuclear dipolar interactions can occur between molecules of the same type (i.e., either PYR-PYR for ^1^H-^1^H or both TFSI-TFSI for ^19^F-^19^F), and heteronuclear dipolar interactions can only occur between different molecules (i.e., PYR-TFSI for ^1^H-^19^F) [28]. The distinction between the treatments of inter- and intramolecular relaxation does not arise from choosing different relaxation models, but rather from the fact that the spectral density function J(ω) must be defined differently for translational and rotational motions [29]. Additionally, chemical shift anisotropy (CSA) influences the relaxation of ^19^F, more specifically at higher frequencies, because CSA relaxation scales with the square of the magnetic field [30]. Thus, CSA also has to be considered to describe the TFSI^−^ relaxation profile. The total relaxation rate fitted with the experimental results is given by Equation (3), and each component of the equation is described below:(3)$$R_{1}^{i}\left(\omega_{i}\right)=R_{1}^{Rot}\left(\omega_{i}\right)+R_{1_{ii}}^{SD}\left(\omega_{i}\right)+R_{1_{ij}}^{SD}\left(\omega_{i}\right)+{\left[R_{1}^{CSA}\left(\omega_{F}\right)\right]}_{i=F}$$

In Equation (3), $R_{1}^{i}\left(\omega_{i}\right)$ is the total longitudinal relaxation rate measured experimentally for nucleus i. This rate can be deconvoluted into contributions from rotational dynamics $R_{1}^{Rot}\left(\omega_{i}\right)$, self-diffusion of like spins $R_{1_{ii}}^{SD}\left(\omega_{i}\right)$ and unlike spins $R_{1_{ij}}^{SD}\left(\omega_{i}\right)$, and, in the case of ^19^F, a contribution from chemical shift anisotropy $R_{1}^{CSA}\left(\omega_{F}\right)$.

For quadrupolar nuclei such as ^7^Li (I = 3/2), the dominant spin-lattice relaxation mechanism is quadrupolar relaxation, which arises from fluctuations in the electric field gradient (EFG) at the nuclear site. These fluctuations interact with the nuclear electric quadrupole moment, leading to an energy exchange with the lattice. In the case of ^7^Li, relaxation primarily reflects the dynamic modulation of the EFG tensor due to both intra- and intermolecular charge distributions in the local environment. In the eutectic LITFSI-TFSI systems, where Li^+^ typically exists as a solvated ion [12], the EFG originates mainly from intermolecular interactions established in the coordination shell. The temporal fluctuations of this EFG are influenced by a combination of long-range translational diffusion and local (short-range) reorientation motions of the Li^+^ ion and its coordinating species [31]. Thus, the total relaxation rate fitted with the experimental results for ^7^Li $R_{1}^{7Li}\left(\omega \right)$ is given by Equation (4):(4)$$R_{1}^{7Li}\left(\omega \right){=R}_{1}^{Q}\left(\omega \right)$$

Here, $R_{1}^{Q}\left(\omega \right)$ is the quadrupolar relaxation mechanism, as described in Section 2.4.4.

The online platform *fitteia^®^* [32,33] was used to fit all relaxation models to the experimental data in this study. This platform allows users to define well-established molecular parameters, including intramolecular and intermolecular distances obtained from molecular dynamics simulations (e.g., *r_H–H_* = 2.3 Å, the shortest possible distance between two proton spins in pyrazole; *d_H–H_* = 3.2 Å; *d_F–F_* = 5.1 Å; *d_H–F_* = 2.9 Å, the intermolecular distances), as well as the calculated spin densities (Appendix A) and the self-diffusion coefficients measured by PFG-NMR (Appendix A). The fitting procedure then determines the correlation times and chemical shift anisotropy as the only output variables. For all parameters requiring optimization, fitting was performed using a non-linear least squares minimization approach. The resulting least-squares minima consistently fell within physically meaningful ranges, aligning well with values reported in prior studies and supported by theoretical models. Further details about *fittea^®^* can be found in the literature [32,33].

#### 2.4.1. Relaxation Due to Intermolecular Contributions ($R_{1_{ii}}^{SD}\;and\;R_{1_{ij}}^{SD}$)

For the intermolecular contributions to relaxation, the Force-Free-Hard-Sphere (FFHS) model was used to model the translational diffusion of the TFSI^−^ anion (^19^F) and the pyrazole (^1^H) cation, with $R_{1\_ii}^{SD}$ representing the co-ion interactions and $R_{1\_ij}^{SD}$ representing the counter-ion interactions [34,35]. The FFHS model mathematically describes translational diffusion as a movement of hard spheres with Fick’s diffusion equation via the Hwang–Freed spectral density function [34,35]. The Hwang–Freed spectral density function (*J*) describes intermolecular spectral density as a function of Larmor angular frequency ($\omega_{i}$) of the nucleus *i* with the characteristic correlation time ($\tau_{\mathrm{SD}}$) [35].

The homonuclear translational diffusion model ($R_{1\_ii}^{SD}$) can be written as Equation (5):(5)$$R_{1\_ii}^{SD}\left(\omega_{i}\right)=\frac{4}{15}\frac{N_{i}\pi }{d_{ii}^{3}}K_{dip}^{ii}\left[J_{SD}^{\left(1\right)}\left(\omega_{i}\right)+J_{SD}^{\left(2\right)}\left(2\omega_{i}\right)\right]$$

Here, the indexes *i* and *j* stand for either H or F, $\omega_{i}=2\pi \nu_{i}$ and $\nu_{i}$ is the Larmor frequency of the nucleus *i* in Hertz; the spectral density function ${J_{SD}^{\left(k\right)}\left(\omega_{i}\right)=c}_{k}J\left(\omega_{i}\right),\;w$ here $c_{k}=6,\;1,\;4$ for k=0, 1, 2, respectively; $d_{ii}$ is the intermolecular distance between like nuclei of type *i*, $N_{i}$ is the spin density (see Appendix A), and $K_{dip}^{ij}$ represents the dipolar coupling constant expressed by Equation (6):(6)$$K_{dip}^{ij}=\frac{3}{2}{\left(\frac{\mu_{0}\gamma_{i}\gamma_{j}\hbar }{4\pi }\right)}^{2}$$

Here, the indexes *i* and *j* stand for either H or F, $\mu_{0}$ is the vacuum magnetic permeability, $\gamma_{k}$ (*k* = *i*, *j*) is the gyromagnetic ratio of nucleus *k*, and ℏ is the reduced Planck constant.

The heteronuclear translational diffusion model ($R_{1\_ij}^{SD}$) can be written as Equation (7):(7)$$R_{1\_ij}^{SD}\left(\omega_{i}\right)=\frac{4}{15}\frac{N_{j}\pi }{d_{ij}^{3}}K_{dip}^{ij}\left[\frac{1}{3}J\left(\left|\omega_{i}-\omega_{j}\right|\right)+J\left(\omega_{i}\right)+2J\left(\omega_{i}+\omega_{j}\right)\right]\;i\neq j$$

Here, the indexes *i* and *j* stand for either H or F, $\omega_{i}=2\pi \nu_{i}$ and $\nu_{i}$ is the Larmor frequency of the nucleus *i* in Hertz, J is the spectral density function, $d_{ij}$ is the intramolecular distance between nuclei of type *i* and type *j*, $N_{j}$ is the spin density (see Appendix A), and $K_{dip}^{ij}$ represents the dipolar coupling constant expressed by Equation (6).

The Hwang–Freed spectral density function is defined as below (Equation (8)):(8)$$J\left(\omega_{i},\tau_{SD}\right)=72\frac{3}{4\pi }\int_{0}^{\infty }\frac{u^{2}}{81+9u^{2}-2u^{4}+u^{6}}\frac{{u^{2}\tau }_{\mathrm{SD}}}{u^{4}+{\left(\omega_{i}\tau_{\mathrm{SD}}\right)}^{2}}du$$
with the characteristic correlation time:(9)τSD=dij22Dij

Here, the indexes *i* and *j* stand for either H or F, $\omega_{i}=2\pi \nu_{i}$, and $\nu_{i}$ is the Larmor frequency of the nucleus *i* in Hertz, $N_{j}$ is the spin density of its corresponding nuclei, calculated as described in the Appendix A, and $d_{ij}$ is the most probable distance between two nuclei (H–H, F–F, H–F, or F–H) belonging to separate molecules, determined by molecular dynamics simulations as described in the SI. More specifically, radial distribution functions (RDFs) were computed for relevant atom pairs (H–H, F–F, and H–F) over the production trajectory, and the peak positions were used to determine the probable intermolecular distances for use in the relaxation models. Additionally, *u* is a dimensionless integration variable, ℏ is Plank’s constant divided by 2π, $\mu_{0}$ is the vacuum magnetic permeability, γ is the gyromagnetic ratio for its corresponding nucleus, and $D_{ij}$ is the relative diffusion coefficient, which is calculated from the sum of the diffusion coefficients obtained by PFG-NMR of the molecules involved (see Appendix A) [34,35].

#### 2.4.2. Relaxation Due to Intramolecular Contributions ($R_{1}^{Rot}$)

Relaxation due to intramolecular contributions is caused by rotational/re-orientational motion of the molecule. In the case of elongated molecules, such as TFSI^−^, $R_{1}^{Rot}$ was treated using Nordio’s model, which considers two distinct correlation times: one along the main molecular axis (*τ_z_*) and another along the short axis (*τ_x_*). Nordio’s rotational model ($R_{1}^{Rot}$) can be written as Equations (10)–(13) [36,37,38]:(10)$$R_{1}^{Rot}\left(\omega_{i}\right)=\;\frac{3}{4}K_{dip}^{ii}\left[J_{R}^{\left(1\right)}\left(\omega_{i}\right)+J_{R}^{\left(2\right)}\left(2\omega_{i}\right)\right]$$
where $K_{dip}^{ii}$ is the dipolar coupling constant in the case of homonuclear interaction (Equation (6)), and $J_{R}^{\left(k\right)}$ (for k = 0, 1, 2) is the spectral density function defined as in Equation (11).

In the case of isotropic liquids, and in the absence of orientational order (*S* = 0), the spectral density function, $J_{R}^{\left(k\right)}$ (for k = 0, 1, 2) can be written as Equation (11) [39]:(11)$$J_{R}^{\left(k\right)}\left(\omega \right)=\;\frac{4}{15}c_{k}\sum_{m=0}^{2}A^{\left(m\right)}\frac{\tau_{m}}{1+\tau_{m}^{2}\omega^{2}}$$
where $c_{k}=6,\;1,\;4$ for k=0, 1, 2, respectively, and(12)$$A^{\left(m\right)}=\left\{\begin{array}{c} \bar{{\left(3\cos^{2}\alpha_{\mathrm{ij}}-1\right)}^{2}/\left({4r}_{\mathrm{ij}}^{6}\right)},m=0\\ \bar{3\sin^{2}2\alpha_{\mathrm{ij}}/\left({4r}_{\mathrm{ij}}^{6}\right)},\;m=1\\ \bar{3\sin^{4}\alpha_{\mathrm{ij}}/\left({4r}_{\mathrm{ij}}^{6}\right)},\;m=2 \end{array}\right.$$

Here, $\alpha_{ij}$ is the angle that each dipole spin vector makes with the main axis, *r_ij_* is the inter spin distance, and m enumerates three distinct angular terms that arise from the second-rank dipolar interaction tensor. $\tau_{m}$ depends on correlation times for rotation around the main (*τ_z_*) and short axis (*τ_x_*), and for *S* = 0 (absence of any orientational order):(13)$${\left(\tau_{m}\right)}^{-1}=\;{\left(6\tau_{z}\right)}^{-1}\left[6+\left(\frac{\tau_{x}}{\tau_{z}}-1\right)m^{2}\right]$$

In the case of nearly spherical molecules, $\tau_{x}$ = $\tau_{z}$ = τ and Equation (11), with the use of Equations (12) and (13), can be simplified to Equation (14):(14)$$J_{R}^{\left(k\right)}\left(\omega \right)=\;\frac{4}{15}c_{k}\frac{\tau }{1+\tau^{2}\omega^{2}}\left\langle \left.\frac{1}{r_{ij}^{6}}\right\rangle \right.$$
which is equivalent to the classical Bloembergen Purcell Pound (BPP) spectral density function [40]. In Equation (14), the indexes *i* and *j* stand for either H or F, ω=2πν and ν is the Larmor frequency in Hertz, $c_{k}$ is a numerical weight for each rank of the dipolar interaction, τ is the rotational correlation time, and $r_{ij}$ is the internuclear distance between nuclei *i* and *j*. The BPP model was used to model the rotational motion of Pyrazole (^1^H), which is a small molecule, and Nordio’s model was used to quantify the rotational motion of the large ion TFSI^−^ (^19^F) [30,37,38,41].

#### 2.4.3. Relaxation Due to Chemical Shift Anisotropy ($R_{1}^{CSA}$)

We accounted for chemical shift anisotropy (CSA) in the spin-lattice relaxation of TFSI^−^ (^19^F), denoted by ($R_{1}^{CSA}$) (Equation (15)) [42]. As previously shown in the literature, CSA influences ^19^F relaxation in small molecules [43], as well as in TFSI^−^ molecules [31,44], especially at higher frequencies, because CSA relaxation scales with the square of the magnetic field, as expressed in Equation (17) [30].(15)$$R_{1}^{CSA}\left(\omega \right)=\frac{6}{40}\omega^{2}{C_{SA}}^{2}\left(10^{-12}\right)\frac{\tau_{F}}{1+{\left(\omega \tau_{F}\right)}^{2}}$$

Here, $C_{SA}$ is the ^19^F chemical shift anisotropy in ppm, ω=2πν and ν is the Larmor frequency in Hertz, and *τ_F_* is the correlation time for the rotation of the CF_3_ group [31].

#### 2.4.4. Relaxation Due to the Quadrupolar Mechanism ($R_{1}^{Q}$)

The local motion of Li^+^ (^7^Li) is dominated by a quadrupolar relaxation mechanism. Li^+^ forms contact ions pairs and large aggregates with the TFSI^−^ anions [45], and the Li^+^ cation hops between anion coordination sites within its first coordination shell [46]. In eutectic solvents, Li^+^ transport can also proceed via a vehicular pathway, in which the cation moves together with its immediate coordination environment [47]. As a first approximation, assuming isotropic stochastic motion that modulates the local electric field gradient with a single correlation time τ_C_ ≈ τ_jump_, the ^7^Li spin-lattice quadrupolar relaxation rate $R_{1}^{Q}$ can be written using Lorentzian spectral densities (Equations (16) and (17)) [28,31]:(16)$$R_{1}^{Q}\left(\omega \right)=\;\frac{3\pi^{2}}{10}\frac{\left(2I+3\right)}{I^{2}\left(2I-1\right)}\left(1+\frac{\eta^{3}}{3}\right)C_{Q}^{2}\left[\frac{\tau_{jump}}{1+{\left(\omega \right)}^{2}{\tau_{jump}}^{2}}+\;\frac{4\tau_{jump}}{1+{\left(2\omega \right)}^{2}{\tau_{jump}}^{2}}\right]$$
with the following jump correlation time:(17)τjump=⟨r⟩26D

In these two equations (Equations (16) and (17)), *I* refers to the nuclear spin quantum number for ^7^Li, *η* is the asymmetry parameter of the electric field gradient, *C_Q_* is the quadrupolar coupling constant, ${\left\langle r\right\rangle }^{2}$ is the average distance covered by one jump within the characteristic local motion, and *D* is the ^7^Li diffusion coefficient (see Appendix A).

### 2.5. Viscosity and Conductivity

The viscosities of the LiTFSI–Pyrazole Eutectic Solvents were measured at 10 °C using a Discovery Hybrid Rheometer-3 (DHR-3, TA Instruments, New Castle, DE, USA) equipped with a Dual Stage Peltier Plate (DSPP) system and 40 mm stainless-steel parallel plates. The gap between the plates was set to 500 µm, and the samples were equilibrated for 300 s at 10 °C before measurement. A run time of 600 s was applied, and viscosity values were recorded using the TRIOS software Version 5.10.

Ionic conductivity was determined using a custom-built four-electrode conductivity cell (316 stainless steel electrodes). Approximately 1.5 mL of liquid sample was injected into the cell and equilibrated for 2 h at 10 °C inside a Tenney Environmental Chamber. Electrochemical Impedance Spectroscopy (EIS) was performed with a BioLogic SP-200 (Bio-Logic Science Instruments, Grenoble, France) potentiostat controlled by EC-Lab software Version 11.60, over the frequency range of 1 Hz–5 MHz. The solution resistance was obtained from the high-frequency intercept of the Nyquist plot, and conductivity (σ) was calculated according to Equation (18):(18)$$\sigma =\frac{L}{RA}$$

Here σ is the conductivity of eutectic solvents (mS/cm), *L* is the length between sensing electrodes (1.7 cm), A is the cross-sectional area available for current flow in the sample (0.22 cm^2^), and R is the resistance from high frequency intercept of EIS (Ω).

## 3. Results and Discussion

### 3.1. Transport Properties

Pulsed Field Gradient (PFG) NMR was employed to examine translational dynamics by measuring the self-diffusion coefficients of the LiTFSI–Pyrazole Type IV Eutectic Solvents in the ^1^H, ^19^F, and ^7^Li domains at 10 °C. The results reveal an intricate interplay between the lithium salt and pyrazole at different concentrations (see Figure 3a and Appendix A). Unusually, in LiTFSI:PYR (1:2), ^19^F diffuses faster than ^1^H, suggesting that the pyrazole molecules experience stronger intermolecular interactions, such as hydrogen bonding or coordination with Li^+^, which hinder their motion more than that of the TFSI^−^ anions. Therefore, the diffusion results suggest that there is a different structural arrangement of the solution for LiTFSI:PYR (1:2) when compared to the other compositions. On the other hand, in the more diluted eutectic solvents LiTFSI:PYR (1:3), (1:4), and (1:5), the ^1^H diffuses faster than the ^19^F, consistent with weaker interactions and a reduced degree of PYR—LiTFSI association. Indeed, in their study, Liu et al. conducted electrochemical measurements and material characterization of various LiTFSI–azole compositions. Using DFT calculations and Raman spectroscopy, they demonstrated that the nitrogen atom in pyrazole can form an asymmetric coordination structure with Li^+^, particularly pronounced at the LiTFSI:PYR (1:2) ratio [12].

Ionic conductivity measurements were performed in order to quantify the transport properties of LiTFSI:PYR Type IV ES in view of their potential application as lithium-ion battery electrolytes. To be comparable with the state-of-the-art organic carbonates-based electrolytes, a functional electrolyte is generally expected to exhibit ionic conductivities in the range of 5 to 15 mS·cm^−1^ at room temperature [48]. According to our results, the LiTFSI:PYR systems have a varying ionic conductivity within the range of 0.6 to 3.13 mS·cm^−1^ at 10 °C (see Figure 3b, Appendix A). Thus, as anticipated, at lower temperatures, the overall slowdown in ion dynamics is reflected in a decrease in macroscopic ionic conductivity, consistent with the temperature dependence of ion transport processes. Notably, these values are comparable to or exceed those reported for lithium salt–based ionic liquid electrolytes at sub-ambient temperatures. It has been shown that pyrrolidinium ionic liquid systems containing LiTFSI typically exhibit conductivities in the 10^−3^–10^−2^ S·cm^−1^ range between −20 and 40 °C, with restricted ion mobility at low temperatures [49,50]. This comparison highlights the benefit of exploring eutectic solvents as promising alternatives to conventional ionic liquids for enhancing low-temperature ion transport.

Looking to Figure 3b, it is apparent that the LiTFSI:PYR (1:2) sample has the lowest ionic conductivity and the LiTFSI:PYR (1:4) sample has the highest. This trend does not directly correlate with the self-diffusion coefficients, which increase as the lithium salt concentration decreases. Instead, the drop in conductivity at high salt concentration likely results from the formation of long-lived ion pairs or contact Li-TFSI ions pairs and ion-ion correlations [51,52], which reduces the number of free charge carriers contributing to ionic transport. In the LiTFSI:PYR (1:2) composition, strong interactions between Li^+^ and pyrazole may lead to structural rearrangements that restrict ion mobility and increase ion–ion correlations. Conversely, the LiTFSI:PYR (1:4) composition may represent a compromise, maintaining sufficient charge density to support bulk ionic transport.

A way to account for the ion correlation is by calculating the ionicity by the inverse Haven ratio [53,54,55]:(19)$$\frac{\Lambda_{EIS}}{\Lambda_{NMR}}$$

Here $\Lambda_{EIS}$ is the molar conductivity of the solvent determined by EIS, and $\Lambda_{NMR}$ is the molar conductivity predicted from self-diffusion coefficients measured by PFG NMR. $\Lambda_{NMR}$ is calculated via the Nernst–Einstein Equation (Equation (20)):(20)$$\Lambda_{NMR}=\frac{F^{2}}{RT}(\nu_{+}z_{+}^{2}D_{{Li}^{+}}+\nu_{-}z_{-}^{2}D_{{TFSI}^{-}})$$

Here *F* is Faraday’s constant, *R* is the gas constant, $z_{i}$ are the ionic charges, $\nu_{i}$ are the stoichiometric numbers, *T* is temperature in Kelvin, and $D_{i}$ is a self-diffusion coefficient for the nucleus *i* as determined using PFG NMR.

For all LiTFSI:PYR eutectic solvents, the ionicity was found to be below 0.1 (see Appendix A), indicating strong correlations between ions that limit the effective conduction of charges [56]. Similar values of low ionicity have previously been reported for eutectic solvents composed of molecular hydrogen bond acceptors and donors (Type V DES) doped with various concentrations of LiTFSI [57]. These findings highlight that, although molecular hydrogen bond donors or acceptors enable LiTFSI dissolution and eutectic formation, they do not significantly enhance Li-salt correlation to support efficient ionic transport.

Viscosity measurements were conducted to complement the transport properties investigation of the LiTFSI–Pyrazole Type IV Eutectic Solvents. For the investigated Type IV eutectic solvents, viscosity values ranged from 0.3 to 0.7 Pa·s at 10 °C (see Figure 3c, Appendix A). These values are notably high when compared to other classes of DES. Traditional Type III DESs (e.g., choline chloride–based systems) typically exhibit viscosities ranging from 0.02 to 0.1 Pa·s at 20 °C [58], which can then increase sharply at lower temperatures due to extensive hydrogen-bond networks. In contrast, LiTFSI-containing Type IV DESs commonly report higher viscosities, from 0.07 to 2.8 Pa·s, for a broad range of temperatures, which shows that changing the HBD is a favorable alternative to tuning their ion transport [59].

As expected, viscosity and conductivity exhibit an inverse relationship, with higher viscosities generally corresponding to lower ionic conductivities [60]. This trend is evident in Figure 3, as the LiTFSI:PYR (1:2) sample, which exhibited the lowest ionic conductivity (see Figure 3b), shows the highest viscosity (see Figure 3c). This inverse relationship supports the interpretation that ion mobility is hindered in more viscous media. The elevated viscosity observed for LiTFSI:PYR (1:2) likely reflects stronger intermolecular interactions between PYR and LiTFSI.

As previously mentioned, the LiTFSI:PYR (1:4) sample exhibits the highest ionic conductivity, consistent with its lower viscosity (see Figure 3b,c). Notably, although the self-diffusion coefficients are not the highest in this composition (e.g., the 1:5 sample shows faster molecular diffusion, as shown in Figure 3a), the conductivity is superior due to its higher ionicity. By contrast, in the more diluted compositions, the reduced charge carrier concentration offsets the faster diffusion of individual species.

### 3.2. Local Dynamics

Figure 4 shows the NMRD profiles of ^1^H, ^19^F, and ^7^Li for the LiTFSI:PYR systems at 10 °C and raw data are reported in Appendix A. At high frequencies (hundreds of MHz), the relaxation rates of the four mixtures converge to nearly the same range, indicating that the fast local dynamics due to rotation/reorientation dominating at this frequency are less sensitive to composition trend. In contrast, at lower frequencies, the profiles significantly diverge, with LiTFSI:PYR (1:2) exhibiting substantially higher R_1_ values than the lower-salt mixtures for all the nuclei (see Figure 4, as well as Equations (10), (15) and (16)). At first glance, this behavior might be attributed to viscosity. However, as shown in Figure 3c, LiTFSI:PYR (1:4) is the least viscous system, yet it does not display the lowest R_1_ values. This indicates that factors beyond viscosity—particularly ion–ion interactions and local coordination dynamics—play a decisive role in governing the nucleus-specific relaxation behavior of the individual components within these eutectic systems. To provide a physically meaningful interpretation of these NMRD profiles, we quantified the underlying relaxation mechanisms.

#### 3.2.1. ^1^H and ^19^F Relaxation Mechanisms

The relaxation of ^1^H nuclei was modeled using a combination of the Force-Free Hard Sphere (FFHS) model for translational diffusion and the Bloembergen-Purcell-Pound (BPP) model for rotational motion (Figure 5). As shown in Figure 5, the translational relaxation mechanism dominates the low-frequency region of the ^1^H relaxation profile and at higher frequencies, the rotational contribution increases, consistent with a fast rotational motion of pyrazole molecules.

Unlike the other compositions, the ^1^H relaxation profile of LiTFSI:PYR (1:2) required a two-term BPP model to achieve a satisfactory fit, indicating the presence of at least two motional environments for PYR. This behavior consists of the coexistence of (i) PYR molecules undergoing relatively free rotation and (ii) PYR involved in Li^+^-associated structures (e.g., asymmetric PYR:LiTFSI complexes), which exhibit slower and differently weighted re-orientational dynamics. This interpretation points to heterogeneity in the PYR motional dynamics, although alternative sources of dynamic heterogeneity (e.g., distributions of correlation times [61]) cannot be excluded.

The ^19^F relaxation data (Figure 6) were modeled using a combination of FFHS for translational diffusion, Nordio’s model to describe the asymmetric rotational motion of TFSI^−^ (accounting for rotation along the main and short axes), and chemical shift anisotropy (CSA). The contribution of CSA is evident from the upturn in R_1_ values at high frequencies, particularly above 300 MHz. The requirement for a second rotational component in the modeling of the ^19^F relaxation data for the LiTFSI:PYR (1:2) solvent may suggest the presence of more complex coordination and associated rotational dynamics in this sample.

The relaxation parameters in Table 1 provide quantitative evidence that links directly to the transport properties. For the 1:2 composition, the presence of two distinct ^1^H correlation times (τ ≈ 10^−10^ and 10^−8^ s) and highly anisotropic ^19^F rotational dynamics (τ_z_ ≈ 10^−10^–10^−7^ s; τ_x_ ≈ 10^−7^–10^−6^ s) point to strong and heterogeneous PYR–LiTFSI associations. These microscopic constraints are consistent with the macroscopic signatures of this sample: the lowest conductivity, the highest viscosity, and ionicity below 0.1. In contrast, the 1:3 and 1:4 systems can be described by single correlation times in the 10^−9^ s range, reflecting more homogeneous dynamics. Notably, in the 1:4 solvent this relatively fast and isotropic TFSI^−^ reorientation correlates with the highest conductivity and lowest viscosity across the series, indicating that it represents an optimal compromise: Li-TFSI interactions are sufficiently weakened to enhance ionic mobility, while the carrier concentration remains high enough to sustain efficient transport. Finally, the 1:5 system shows even faster local motions (τ ~ 10^−10^–10^−8^ s), but its lower carrier concentration offsets the dynamic advantage, leading to a reduction in conductivity.

#### 3.2.2. ^7^Li Relaxation Mechanisms

The ^7^Li relaxation profiles (Figure 7) were modeled using the quadrupolar relaxation model, which accounts for jump dynamics of Li^+^ within its solvation shell [28,31]. The best fits were obtained by incorporating parameters for quadrupolar coupling and jump diffusion, which reflect lithium’s dynamic exchange between TFSI^−^ coordination environments.

Quadrupolar coupling constants (*C_Q_*) were obtained for ^7^Li through fitting our NMRD relaxation profiles (see Figure 7, Equations (16) and (17)). In our system, the extracted *C_Q_* values (34–52 kHz) (see Table 2) are fully consistent with previously reported ranges for solvated Li^+^ in liquid electrolytes where Li^+^ experiences coordination environments [62,63]. For comparison, *C_Q_* values found for various Li containing materials ranged from 40 to 110 kHz [64]. The magnitude observed here therefore reflects the non-negligible EFG arising from Li^+^ coordination environment. Thus, while motional averaging suppresses static quadrupolar effects in the observed NMR spectra, the remaining/residual quadrupolar coupling is sufficiently large to be considered as a primary relaxation mechanism for this nucleus.

Numerically, *C_Q_* decreases systematically with dilution, from 52 kHz in LiTFSI:PYR (1:2) to 34 kHz in LiTFSI:PYR (1:5). This reduction reflects a progressive weakening of the local electric field gradient at the Li^+^ site, consistent with a shift from strongly coordinated PYR-LiTFSI environments in the 1:2 system to more dynamic, less ordered coordination at higher PYR contents [12].

The fitted effective one-jump distances further support this picture. For 1:2, *r*_1_ = 3.8 Å is significantly shorter than in the other compositions (≈4.1–4.9 Å), indicating closer Li–ligand contacts. At the same time, *r*_2_ is also shorter (0.9 Å vs. 1.6–1.7 Å), suggesting that additional couplings from nearby nuclei contribute more strongly to Li^+^ relaxation in the concentrated sample. Together, these parameters reinforce the interpretation that Li^+^ in 1:2 experiences stronger, more localized coordination, leading to restricted mobility and enhanced relaxation rates.

At higher PYR contents (1:3–1:5), both *C_Q_* and *r*-values indicate weaker and more spatially extended coordination environments, consistent with faster translational dynamics and improved ionic conductivity (Figure 3b). In particular, the 1:4 composition, which combines reduced *C_Q_* (39 kHz) with intermediate Li–ligand distances (*r*_1_ = 4.3 Å, *r*_2_ = 1.6 Å), corresponds to the composition where the balance between charge carrier concentration and Li^+^ mobility yields the highest conductivity.

## 4. Conclusions

This study provides a comprehensive molecular-level analysis of LiTFSI–Pyrazole eutectic solvents using advanced NMR techniques. PFG NMR revealed that ion diffusion is highly dependent on LiTFSI concentration, with the LiTFSI:PYR (1:2) solvent showing unusually low self-diffusion of ^1^H, consistent with increased interactions between Li^+^ and pyrazole. Conductivity and viscosity measurements further supported these findings, demonstrating an inverse relationship with each other, while also confirming the formation of long-lived ion pairs. FFC NMR and subsequent model-fitting clarified the mechanisms underlying these trends, indicating that molecular mobility is governed by a combination of translational diffusion and rotational dynamics, with the LiTFSI:PYR (1:2) solvent displaying enhanced Li^+^–pyrazole coordination. While these results confirm strong molecular interactions and structural organization, they also reveal a significant limitation: the ionic conductivity and ionicity of these eutectic solvents is far below the threshold typically required for practical lithium-ion battery electrolytes. This suggests that, despite favorable molecular coordination, further optimization is necessary to improve conductivity before these materials can be considered viable candidates for battery systems. Ultimately, this work deepens our understanding of ion dynamics in azole-based Type IV DESs and underscores both their promise and current limitations as green, safe, and efficient electrolytes for next-generation lithium-ion batteries. Future studies will investigate the temperature-dependent transport in these LiTFSI–pyrazole eutectic solvents to determine activation energies and evaluate electrolyte performance under conditions relevant to high-temperature energy storage.

## Figures and Tables

**Figure 1 materials-18-05184-f001:**
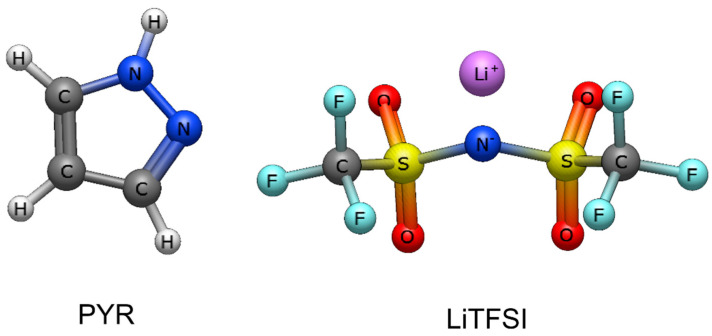
Chemical structure of the components PYR and LiTFSI.

**Figure 2 materials-18-05184-f002:**
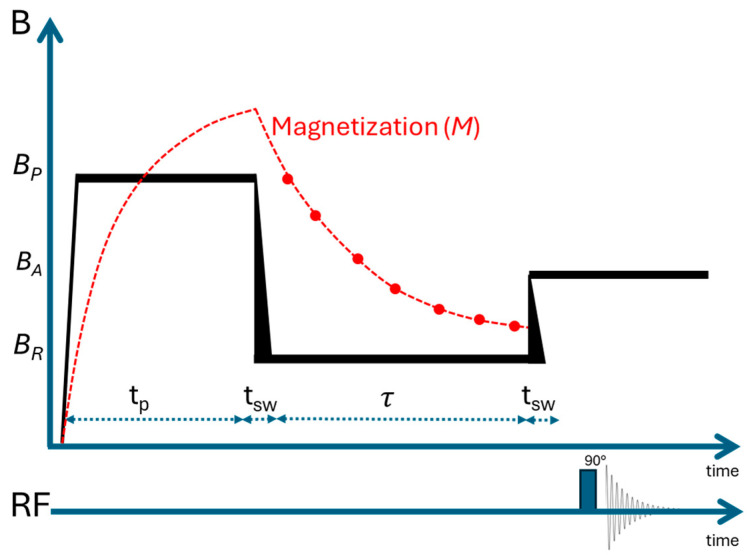
Schematic representation of the FFC-NMR experimental set-up. *B_P_*, *B_A_*, and *B_R_* correspond to the polarization, acquisition, and relaxation fields, respectively, while t_p_ is the polarization time, t_sw_ is the switching time, and *τ* is the interval during which the magnetization evolves at the relaxation field (*B_R_*).

**Figure 3 materials-18-05184-f003:**
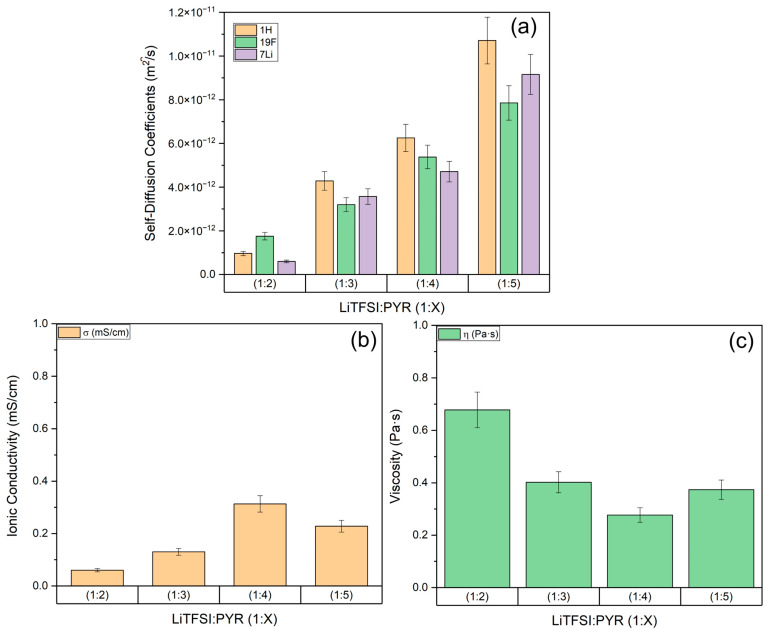
(**a**) Self-diffusion coefficients of each nucleus (^1^H, ^19^F, ^7^Li) at 10 °C (**b**) ionic conductivity at 10 °C (**c**) viscosity in the LiTFSI:PYR (1:X) ESs at 10 °C. All measurements have a uncertainty of 10%.

**Figure 4 materials-18-05184-f004:**
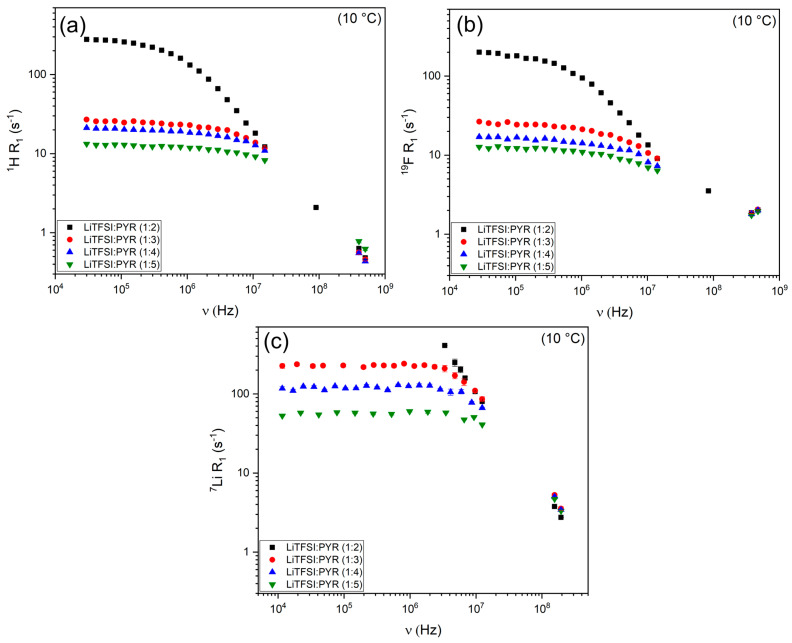
R_1_ NMRD profiles at 10 °C for (**a**) ^1^H, (**b**) ^19^F and (**c**) ^7^Li in LiTFSI:PYR eutectic mixtures as a function of composition.

**Figure 5 materials-18-05184-f005:**
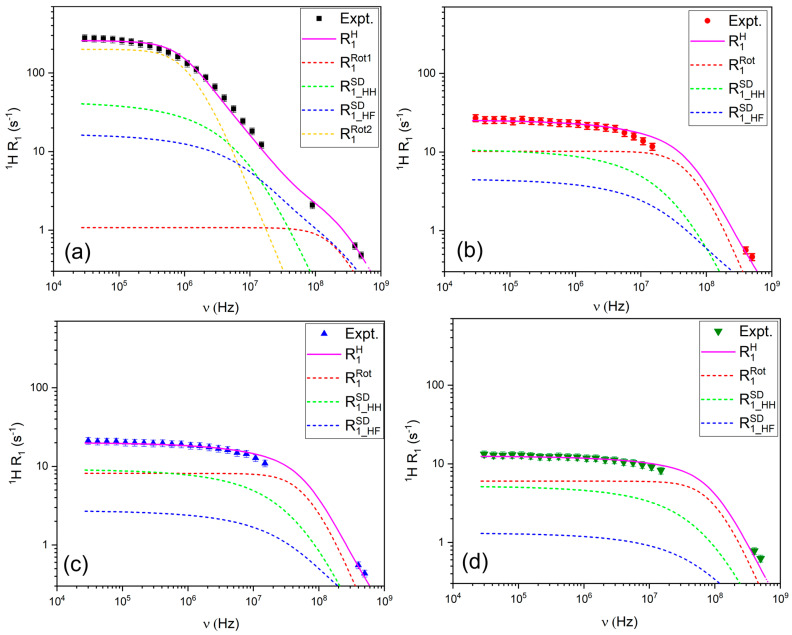
R_1_ relaxation profiles for ^1^H in (**a**) LiTFSI:PYR (1:2), (**b**) LiTFSI:PYR (1:3), (**c**) LiTFSI:PYR (1:4), and (**d**) LiTFSI:PYR (1:5) solvents as a function of composition and their corresponding fitting.

**Figure 6 materials-18-05184-f006:**
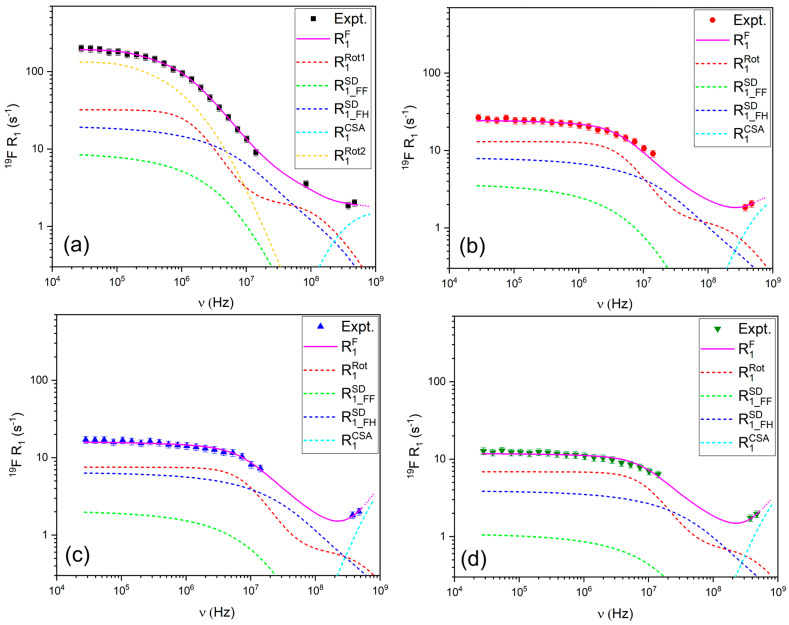
R_1_ relaxation profiles for ^19^F in (**a**) LiTFSI:PYR (1:2), (**b**) LiTFSI:PYR (1:3), (**c**) LiTFSI:PYR (1:4), and (**d**) LiTFSI:PYR (1:5) solvents as a function of composition and their corresponding fitting.

**Figure 7 materials-18-05184-f007:**
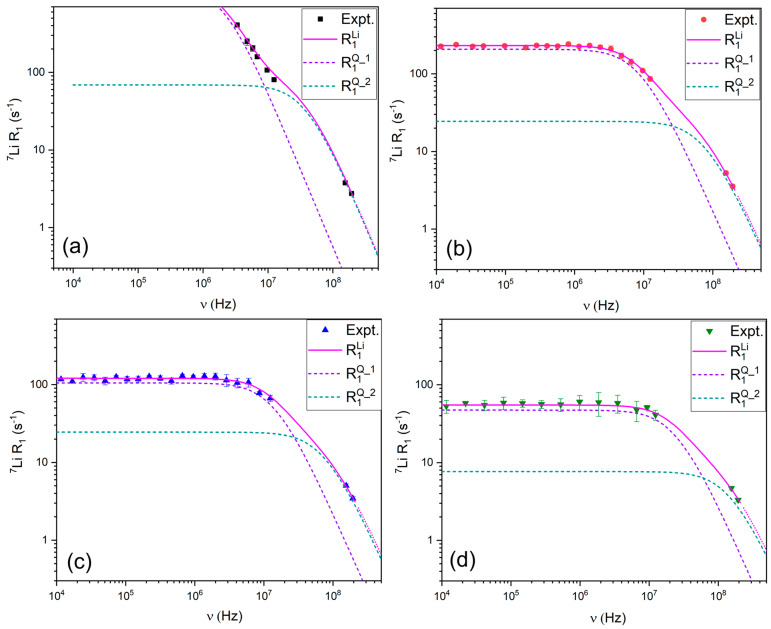
R_1_ relaxation profiles for ^7^Li in (**a**) LiTFSI:PYR (1:2), (**b**) LiTFSI:PYR (1:3), (**c**) LiTFSI:PYR (1:4), and (**d**) LiTFSI:PYR (1:5) solvents as a function of composition and their corresponding fitting.

**Table 1 materials-18-05184-t001:** ^1^H and ^19^F fitting parameters for the investigated systems.

	LiTFSI:PYR (1:2)	LiTFSI:PYR (1:3)	LiTFSI:PYR (1:4)	LiTFSI:PYR (1:5)
**Fitting** **Parameters**	**^1^H**	**^1^H**	**^1^H**	**^1^H**
*τ* (s)	4.0 ± 0.1 × 10^−10^ 7.9 ± 0.4 × 10^−8^ (w = 0.48 ± 0.05)	1.6 ± 0.2 × 10^−9^	1.3 ± 0.2 × 10^−9^	9.4 ± 0.2 × 10^−10^
**Fitting** **Parameters**	**^19^F**	**^19^F**	**^19^F**	**^19^F**
*τ_z_* (s)	3.4 ± 0.2 × 10^−9^ 6.1 ± 0.1 × 10^−7^ (w = 0.20 ± 0.02)	1.6 ± 0.1 × 10^−9^	6.9 ± 0.1 × 10^−10^	9.5 ± 0.1 × 10^−10^
*τ_x_* (s)	2.9 ± 0.1 × 10^−7^ 2.2 ± 0.1 × 10^−6^ (w = 0.20 ± 0.02)	9.3 ± 0.4 × 10^−8^	5.4 ± 0.3 × 10^−8^	4.8 ± 0.1 × 10^−8^
*τ_F_* (s)	5.7 ± 0.1 × 10^−10^	2.7 ± 0.2 × 10^−10^	1.4 ± 0.1 × 10^−10^	1.6 ± 0.2 × 10^−10^
*C_SA_* (ppm)	78 ± 5	75 ± 5	90 ± 6	83 ± 4

**Table 2 materials-18-05184-t002:** ^7^Li fitting parameters for the investigated systems.

Fitting Parameters	LiTFSI:PYR (1:2)	LiTFSI:PYR (1:3)	LiTFSI:PYR (1:4)	LiTFSI:PYR (1:5)
*C_Q_* (kHz)	52 ± 1	44 ± 1	39 ± 1	34 ± 1
*r*_1_ (Å)	3.8 ± 0.02	4.9 ± 0.1	4.3 ± 0.1	4.1 ± 0.1
*r*_2_ (Å)	0.9 ± 0.1	1.7 ± 0.1	1.6 ± 0.1	1.6 ± 0.1

## Data Availability

The original contributions presented in this study are included in the article/Appendix A. Further inquiries can be directed to the corresponding author.

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
