# Peer review of "Nuclear Magnetic Resonance Dynamics of LiTFSI–Pyrazole Eutectic Solvents"

_materials, 2025, doi:10.3390/ma18225184_

Round 1
Reviewer 1 Report
Comments and Suggestions for Authors
Pelegano-Titmuss et al. present a study on the FFC NMR characterization and diffusion behavior of LiTFSI–pyrazole eutectic solvents. Based on relaxation and self-diffusion measurements, the authors aim to elucidate Li⁺–pyrazole interactions for different ion pairs through modeling of the relaxation profiles.
While the work is conceptually interesting and the study design demonstrates novelty, the overall presentation has a somewhat “textbook” character. The authors rely heavily on fitting models rather than providing in-depth interpretation of the relaxation data and a clearer description of the methodological rationale. Furthermore, certain sections would benefit from a more critical discussion connecting the findings to broader scientific implications and existing literature.
In its current form, the manuscript may not sufficiently engage a broader audience. However, with substantial revisions—particularly through improved theoretical context, clearer interpretation, and emphasis on the novelty and scientific advancement of the work—the study could make a meaningful contribution to the field.
Does the introduction provide sufficient background and include all relevant references?
The introduction is clearly written and provides explicit background coverage with recent references. However, the referencing style needs improvement, as it currently mixes different formats.
The following specific points should also be addressed:
- Line 50: Please expand the abbreviation SEI upon first use.
- Line [Flammability]: Flammability is generally related to dendrite formation—please comment on this relationship.
- Line 152: Why was the ^7Li frequency set to 12 MHz? Please clarify.
- Line 158: Do you mean a permanent magnet here? Please specify.
Is the research design appropriate? Are the methods adequately described?
Do the equation 2 is modified from Bloch equation?
static quadrupolar coupling is effectively averaged to zero by the rapid, random motion (tumbling and diffusion) of the solvated lithium ions.Do you think strong Q coupling in your liquid system?
L202 Hoq do you measure shortest possible distance ?
Line 199 : citation require for fitting platform.
Why do you follow different model relaxation studies for inter / itra molecular motion?
Why CSA is smaller for 1:4 than 1:5?
Are the results clearly presented? Are the conclusions supported by the results?
The Results and Discussion section requires a more deliberate and focused approach to effectively convey the key findings and their broader significance to the scientific community. The Conclusion, however, is clearly written, well supported by the presented results, and effectively summarizes the main outcomes of the study.
Equation 2: Is Equation 2 a modified form of the Bloch equation? Please clarify.
Quadrupolar coupling: The manuscript mentions that the static quadrupolar coupling is effectively averaged to zero by the rapid, random motion (tumbling and diffusion) of solvated lithium ions. Do you expect strong quadrupolar coupling effects in your liquid system under these conditions?
Line 199: A citation is required for the fitting platform used in the analysis.
Line 202: How did you determine the shortest possible distance between interacting species? Please provide methodological details or references.
Why did you choose different relaxation models to describe inter- and intramolecular motions? A brief justification would help clarify this distinction.
Why is the chemical shift anisotropy (CSA) smaller for the 1:4 system compared to the 1:5 composition? Please elaborate on the underlying physical or structural reasoning.
How did you measure CQ not showing any NMR spectra? Why is the Cq value is too much for the solution?
Are all figures and tables clear and well-presented?
Agreed.

Author Response
Reviewer 1
Pelegano-Titmuss et al. present a study on the FFC NMR characterization and diffusion behavior of LiTFSI–pyrazole eutectic solvents. Based on relaxation and self-diffusion measurements, the authors aim to elucidate Li⁺–pyrazole interactions for different ion pairs through modeling of the relaxation profiles.
While the work is conceptually interesting and the study design demonstrates novelty, the overall presentation has a somewhat “textbook” character. The authors rely heavily on fitting models rather than providing in-depth interpretation of the relaxation data and a clearer description of the methodological rationale. Furthermore, certain sections would benefit from a more critical discussion connecting the findings to broader scientific implications and existing literature.
In its current form, the manuscript may not sufficiently engage a broader audience. However, with substantial revisions—particularly through improved theoretical context, clearer interpretation, and emphasis on the novelty and scientific advancement of the work—the study could make a meaningful contribution to the field.
The introduction is clearly written and provides explicit background coverage with recent references. However, the referencing style needs improvement, as it currently mixes different formats.
R: We thank the reviewer for this important comment and for their positive feedback on our introduction. In response, we have revised the referencing style throughout the manuscript to adhere strictly to the Materials (MDPI) formatting guidelines.
Line 50: Please expand the abbreviation SEI upon first use.
R: We thank the reviewer for pointing this out. This line has been amended to make clear that the abbreviation SEI stands for Solid Electrolyte Interphase (Line 50).
Flammability is generally related to dendrite formation—please comment on this relationship.
R: We thank the reviewer for this comment. We agree that dendrite formation in lithium metal batteries can contribute to safety hazards, including short-circuiting that may trigger thermal runaway and, consequently, flammability. However, we would like to note that the flammability concerns discussed in our manuscript refer primarily to the use of flammable organic carbonate solvents in conventional Li-ion electrolytes (EC/EMC, EC/DMC), rather than to dendrite-induced failure mechanisms.
The present study focuses on Li-ion battery electrolytes (not Li-metal systems), where dendrite formation is not the dominant safety limitation. Instead, our motivation for eutectic solvent development is to mitigate the intrinsic flammability and thermal instability of current carbonate-based electrolytes. We have clarified this distinction in the revised manuscript to avoid potential ambiguity (Lines 59-62).
Line 152: Why was the 7Li frequency set to 12 MHz? Please clarify.
R: We thank the reviewer for raising this important question. The frequencies in which we conducted 7Li FFC was capped at 12 MHz in 7Li Larmor frequency, which is equivalent to 35 MHz in 1H Larmor frequency. The theoretical limit of our instrumentation is 40 MHz in 1H Larmor frequency, so we set the maximum 7Li frequency to 35 MHz in 1H Larmor frequency as a safety precaution. We have added this explanation to the manuscript, and have now made it clear in the text which frequencies are in which nucleus Larmor frequency (Lines 165-168).
Line 158: Do you mean a permanent magnet here? Please specify.
R: We thank the reviewer for this helpful suggestion. The 300, 400, and 500 MHz NMR spectrometers that we used are fixed-field spectrometers, and the 90 MHz NMR spectrometer is a permament magnet. We made this difference clear in the manuscript. Lines 182-185.
The Results and Discussion section requires a more deliberate and focused approach to effectively convey the key findings and their broader significance to the scientific community. The Conclusion, however, is clearly written, well supported by the presented results, and effectively summarizes the main outcomes of the study.
Equation 2: Is Equation 2 a modified form of the Bloch equation? Please clarify.
R: We thank the reviewer for bringing up this important question. In short, yes, Equation 2 is based on Bloch’s equation for the z component of magnetization (page 19 of Kimmich, R. Field-Cycling NMR Relaxometry; Royal Society of Chemistry: Cambridge, 2023). However, in Equation 2, the longitudinal magnetization (Mz) decays from M0, which is the Curie magnetization (see word file attached), in the polarization field BP. So, Equation 2 is derived from both the Block equation and Curie’s law for magnetization. Curie’s law for magnetization defines M0 through the characterization of electron spins in thermal equilibrium at high temperatures (see page 17 of Kimmich, R. Field-Cycling NMR Relaxometry; Royal Society of Chemistry: Cambridge, 20):
Here, the experimental variables are the absolute temperature T, and the external quantizing flux density B0. Additionally, n is the number of spins per unit volume, is the gyromagnetic constant, is the reduced Planck constant, I is the nuclear spin quantum number, and kB is the Boltzmann constant.
We have clarified this point in the revised manuscript lines 174-176 and Figure 2.
Kimmich, R. Field-Cycling NMR Relaxometry; Royal Society of Chemistry: Cambridge, UK, 2023; Chapter 1: Principle, Purpose and Pitfalls of Field-Cycling NMR Relaxometry.
Quadrupolar coupling: The manuscript mentions that the static quadrupolar coupling is effectively averaged to zero by the rapid, random motion (tumbling and diffusion) of solvated lithium ions. Do you expect strong quadrupolar coupling effects in your liquid system under these conditions?
R: We thank the reviewer for this insightful comment. In liquid electrolytes, rapid isotropic motion of solvated Li⁺ ions does indeed average out the anisotropic components of the electric field gradient (EFG), substantially reducing quadrupolar relaxation effects compared to solids. However, motional averaging does not eliminate the quadrupolar interaction entirely, but reduces it to a residual quadrupolar coupling constant, which can still be significant in electrolytes with strong Li⁺ coordination.
In our system, the extracted CQ values (34–52 kHz) are fully consistent with previously reported ranges for solvated Li⁺ in liquid electrolytes where Li⁺ experiences strong, asymmetric coordination environments. For comparison, Li⁺ in highly coordinated environments such LiC6 exhibits a CQ value of 52 kHz (Marinos et al. 1983- see Ref 63). The magnitude observed here therefore reflects the non-negligible EFG arising from Li⁺ coordination rather than the static coupling expected in solid-state materials. (Lines 497-506)
Line 199: A citation is required for the fitting platform used in the analysis.
R: We thank the reviewer for this suggestion. A citation [Ref 32,33] has been added to the now line 231 where we first introduced the fitting platform that we used in our analysis.
Line 202: How did you determine the shortest possible distance between interacting species? Please provide methodological details or references.
R: We thank the reviewer for this thoughtful suggestion. The shortest possible distances between interacting species (H-H, F-F, and H-F) were calculated using molecular dynamics (MD) simulations (see Supplemetary Information section titled “Determination of the intermolecular and intramolecular distances by MD simulations”). MD simulations were performed for each LiTFSI:PYR composition using LAMMPS with force-field parameters and simulation protocols described in the Supplementary Information. The shortest possible distances were extracted by evaluating the peak positions of the corresponding radial distribution functions (RDFs), which reflect the most probable intermolecular separation distances between nuclei belonging to different molecules. We have added this clarification to Section 2.4.1 lines 271-275 in the manuscript for completeness.
Why did you choose different relaxation models to describe inter- and intramolecular motions? A brief justification would help clarify this distinction.
R: We thank the reviewer for this question. We would like to clarify that the distinction between the treatments of inter- and intramolecular relaxation does not arise from the fact that the spectral density function J(ω) must be defined differently for translational and rotational motions and that in both case they follow the macroscopic equation for spin-lattice relaxation for like spins (page 291 Eq. 76 of Abragam 1961) and unlike spins (page 295 Eq. 87 of Abragam 1961).
Intermolecular relaxation originates from translational diffusion of molecules, where relative motion between spins modulates the dipolar interaction. In this case, one of the possibles appropriate form of J(ω) is given by the Hwang–Freed (Force-Free Hard-Sphere) model, which describes the frequency dependence of dipolar fluctuations caused by diffusion of molecules through space.
In contrast, intramolecular relaxation arises from rotational (re-orientational) motion of rigid or semi-rigid groups within a molecule. Here, J(ω) can follow the Bloembergen–Purcell–Pound (BPP) relaxation model in the case of short and symmetrical molecules or Nordio model for enlogated molecules which requires different correlation times to describe to rotation around the main and short axisi. In both cases dipolar fluctuations are governed by rotational correlation times rather than translational self-diffusion.
We have clarified this point in the revised manuscript Lines 244-252; 280-284; 294-295.
Abragam, A. Principles of Nuclear Magnetism; Oxford University Press: Oxford, UK, 1961.
Why is the chemical shift anisotropy (CSA) smaller for the 1:4 system compared to the 1:5 composition? Please elaborate on the underlying physical or structural reasoning.
R: We thank the reviewer for this insightful comment. After revisiting our analysis, we confirmed that the CSA values for the 1:4 and 1:5 compositions are 90 ± 6 ppm and 83 ± 4 ppm, respectively. Given the overlapping uncertainty ranges, this difference is not statistically significant, and therefore does not warrant interpretation as a meaningful structural or dynamical change in the system.
How did you measure CQ not showing any NMR spectra? Why is the CQ value too much for solution? It is kind of weird.
R: We thank the reviewer for this important question. In this work, the quadrupolar coupling constant (CQ) for ⁷Li was obtained through the fitting of the NMRD relaxation profiles (see Equations 16-17), rather than from NMR spectra. Because our study focuses on field-cycling NMR relaxometry, CQ was extracted as the fitting parameter that best reproduces the experimental R1 dispersion curves over the full frequency range.
We would like to clarify why CQ can be extracted from NMRD even though quadrupolar splittings are not observed in the spectra. In high-field solution NMR, the fast motion of solvated Li⁺ averages the quadrupolar interaction almost completely, so no splitting is detected. However, the quadrupolar interaction does not vanish; it remains as a time-dependent fluctuating interaction and therefore persists as an efficient relaxation mechanism. For this reason, quadrupolar coupling can be quantified through relaxation modeling even when it is invisible in the high-field spectra.
Regarding the magnitude of the extracted CQ values, we note that the reported values represent the residual quadrupolar coupling after motional averaging in solution, not the static solid-state coupling (see response above about quadrupolar coupling). While rapid isotropic motion of solvated Li⁺ ions strongly reduces the effective EFG at the nucleus, residual quadrupolar interactions can remain significant in liquid electrolytes when Li⁺ experiences strongly coordinating solvation environments.
Reviewer 2 Report
Comments and Suggestions for Authors
The paper reports interesting and sound data and interpretation. However, some expressions should be revised. Moreover, I would like to recommend avoiding presenting the interpretation of results before the data themselves. Some suggestions for improvements are the following:
- the sentence on lines 83-85 is not correct. Ionic liquids can have only one cation
- lines 108-110: the paper can contribute to providing a comprehensive understanding of the ion dynamics in Type IV ESs, but it is not a comprehensive work on all Type IV DES.
- explain why you used the particular models for the fits (section 2.4)
- why was the 10°C temperature chosen for measurements?
- Section 3: it seems that the discussion of the data is proposed before the presentation of the data.
Author Response
Reviewer 2
The paper reports interesting and sound data and interpretation. However, some expressions should be revised. Moreover, I would like to recommend avoiding presenting the interpretation of results before the data themselves. Some suggestions for improvements are the following:
The sentence on lines 83-85 is not correct. Ionic liquids can have only one cation.
R: We thank the reviewer for this important comment. We have revised our statement so that, instead of saying that an ionic liquid can have more than one cation, we now state that “ ionic liquid electrolytes doped with lithium salt can have at least two cations (the ionic liquid cation and the Li+).” Lines 86-87.
Lines 108-110: The paper can contribute to providing a comprehensive understanding of the ion dynamics in Type IV ESs, but it is not a comprehensive work on all Type IV DES.
R: We thank the reviewer for pointing this out. We absolutely agree, this manuscript is not contributing to a comprehensive understanding of the ion dynamics of all Type IV eutectic solvents, but only the ion dynamics of LiTFSI-Pyrazole eutectic solvents. We have changed our statement to reflect this, and we now state that “this work delivers a comprehensive understanding of ion motion in LiTFSI-Pyrazole eutectic solvents.” Line 121.
Explain why you used the particular models for the fits (Section 2.4).
R: We thank the reviewer for this question. We have clarify this point on the main manuscript Lines 218-223, 244-252; 280-284; 294-295.
The models were chosen to suitably describe the dynamics of inter- and intramolecular relaxation in case of nuclei ½ (1H and 19F), using the properly density function J(ω) to describe translational and rotational motions, that in both case they follow the macroscopic equation for spin-lattice relaxation for like spins and unlike spins.
Intermolecular relaxation originates from translational diffusion of molecules, where relative motion between spins modulates the dipolar interaction. In this case, one of the possibles appropriate form of J(ω) is given by the Hwang–Freed (Force-Free Hard-Sphere) model, which describes the frequency dependence of dipolar fluctuations caused by diffusion of molecules through space.
In contrast, intramolecular relaxation arises from rotational (re-orientational) motion of rigid or semi-rigid groups within a molecule. Here, J(ω) can follow the Bloembergen–Purcell–Pound (BPP) relaxation model in the case of short and symmetrical molecules or Nordio model for enlogated molecules which requires different correlation times to describe to rotation around the main and short axisi. In both cases dipolar fluctuations are governed by rotational correlation times rather than translational self-diffusion.
For quadrupolar nuclei, such as 7Li (I = 3/2), the dominant spin-lattice relaxation mechanism is quadrupolar relaxation, which arises from fluctuations in the electric field gradient (EFG) at the nuclear site. These fluctuations interact with the nuclear electric quadrupole moment, leading to an energy exchange with the lattice. In the case of 7Li, relaxation primarily reflects the dynamic modulation of the EFG tensor due to both intra- and intermolecular charge distributions in the local environment.
Why was the 10°C temperature chosen for measurements?
R: We thank the reviewer for raising this important point. All measurements were performed at 10 °C because the dispersion profiles are significantly more pronounced at lower temperature, allowing for a more reliable extraction of the characteristic correlation times associated with each relaxation mechanism. In addition, eutectic electrolytes are often considered for operation in low-temperature environments, making transport characterization at reduced temperatures relevant for practical applications. We have now clarified this rationale for our choice of temperature in the revised manuscript line 155-156.
Section 3: It seems that the discussion of the data is proposed before the presentation of the data.
R: We thank the reviewer for this important comment. We have rearranged certain sentences in Section 3, lines 343-354, 363-373, 405-412, in order to ensure that any discussion of data is introduced only after the data itself is presented. However, because Section 3 is a combined section of results and discussions, this section still follows a format in which data is presented and then immediately discussed. We believe that this format enhances clarity for the reader on the significance of our results.
Reviewer 3 Report
Comments and Suggestions for Authors
This work needs improvement in the following areas.
- The introduction looks too elaborate and generalised without any justification regarding the need for this research. Relevant and recent citations must be included. Cite: Polymers 2025, 17, 1758.
- Materials and Methods: For equations (1) to (18), the proper explanation for the terminologies is required. Currently, this is fulfilled partially.
- Discussion for the conductivity and viscosity parameters and findings from those data must be clarified for the readers. Currently, this is incomplete.
- Tables in the supporting information (Tables S1-S7) are not properly cited and discussed. A clarified discussion may boost the readability of this paper.
- Mention the number of data points for the Figure 2 error bar containing plots in their caption.
- The inside text of plots in Figures 4-6 is not in a readable fashion. I suggest improving the text font size as well as the resolution of the Figures.
- Future scope is missing in the conclusion section.
- Replacement of old references with recent and relevant literature is required.
A moderate English polishing and spell check is required.
Author Response
Quality of English Language: The English could be improved to more clearly express the research.
R: We thank the reviewer for this constructive comment. In order to improve the writing in our manuscript, we have completed a spelling and grammar check, and we have improved the rhetoric in select places. All changes made are highlighted in yellow.
The introduction looks too elaborate and generalised without any justification regarding the need for this research. Relevant and recent citations must be included. Cite: Polymers 2025, 17, 1758.
R: We thank the reviewer for this constructive suggestion. Following this recommendation, we have revised the introduction to more clearly highlight the motivation of this study, specifically the need to understand molecular dynamics in deep/eutectic solvent systems, which play a central role in ion transport, viscosity, and electrochemical performance. We now explicitly describe how Fast-Field Cycling NMR provides access to relaxation mechanisms at the nuclear level, enabling the identification of slow and fast motions that cannot be resolved by conventional spectroscopic methods.
We have also incorporated a recent and relevant citation concerning ionic liquids and deep/eutectic solvents with comparable physicochemical properties and relaxation behavior [Ref 10]. This addition helps to better contextualize the current state of the field and clarify the scientific relevance of our approach.
Regarding the specific suggestion to cite Polymers 2025, 17, 1758: we appreciate the reviewer’s recommendation and have now included this citation [Ref 16] in the context of discussing complementary electrolyte systems. However, we note that the cited article investigates ion-conducting copolymer electrolytes (PVDF-HFP/PVP) for solid-state lithium batteries and focuses primarily on dielectric relaxation. In contrast, our work examines liquid eutectic electrolytes and probes nuclear spin relaxation using Fast-Field Cycling NMR, which is governed by translational and rotational dynamics at the molecular scale. Thus, although the citation offers a valuable comparison to alternative electrolyte materials, the physical mechanisms and analytical observables differ substantially from those of polymer-based solid electrolytes.
Materials and Methods: For equations (1) to (18), the proper explanation for the terminologies is required. Currently, this is fulfilled partially.
R: We thank the reviewer for this suggestion and we have provided additional details to the descriptions for each equation (1) through (18). These additional details ensure that the variables in each of the equations are described in the lines that directly follow. Additionally, for further clarity, we have also added Figure 2 to describe the relationship between the Fast Field Cycling methodology and its corresponding equation (Equation 2) .
Discussion for the conductivity and viscosity parameters and findings from those data must be clarified for the readers. Currently, this is incomplete.
R: We thank the reviewer for this comment and have addressed the reviewer’s concern about the clarity of our discussion on conductivity and viscosity parameters. We have done so by adding a comprehensive, numerical-based description of our conductivity and viscosity data, as well as a comparison of our data to the available data in the literature for similar systems. Lines 343-354, 363-373, 405-412.
Tables in the supporting information (Tables S1-S7) are not properly cited and discussed. A clarified discussion may boost the readability of this paper.
R: We thank the reviewer for this helpful suggestion. In response, we have incorporated additional citations to the relevant Supporting Information tables (Tables S1–S7) at appropriate points in the manuscript. These revisions enhance clarity and allow readers to readily access the raw data corresponding to each figure.
Mention the number of data points for the Figure 2 error bar containing plots in their caption.
R: We thank the reviewer for this constructive comment. We have now reported the measurement uncerntainty for our self-diffusion, conductivity, and viscosity measurements in the caption for Figure 3 (Figure 2 has become Figure 3 because a new figure was introduced for clarity in a previous revision).
The inside text of plots in Figures 4-6 is not in a readable fashion. I suggest improving the text font size as well as the resolution of the Figures.
R: We thank the reviewer for this helpful suggestion. We have updated Figures 5-7 (Figures 4-6 have become Figures 5-7 because a new figure, Figure 2, was introduced for clarity in a previous revision). Now, the text font size is larger and the quality of the images is as high as possible for enhanced readability.
Future scope is missing in the conclusion section.
R: We thank the reviewer for pointing this out. We have now added a statement about our future scope in the conclusions. In essence, based on our findings, our future direction is to explore temperature-dependent behaviors of these systems in order to better understand their applications in high temperature energy storage systems. Lines 453-456.
Replacement of old references with recent and relevant literature is required.
R: We thank the reviewer for this suggestion. We have added more recent, relevant references to the manuscript, [Ref 10, 50-51, 60-62, 64-65] in order to keep our work updated according to recent advances and the current state of studies of eutectic solvents electrolytes for storage materials, and expanding for ion-conducting copolymer electrolytes (PVDF-HFP/PVP) for solid-state lithium batteries polymer electrolytes [Ref. 16] as suggested by the reviewer.
Round 2
Reviewer 1 Report
Comments and Suggestions for Authors
Thanks for addressing all the key points according to the revision. I am now ok with the correction and look forward to reading it online.
Reviewer 2 Report
Comments and Suggestions for Authors
The paper has been sufficiently improved and can be published in the present version.
Reviewer 3 Report
Comments and Suggestions for Authors
I appreciate the author's effort to improve the manuscript. I recommend its publication.